# A Potential Predictive Role of the Scalp Microbiome Profiling in Patients with Alopecia Areata: *Staphylococcus caprae*, *Corynebacterium*, and *Cutibacterium* Species

**DOI:** 10.3390/microorganisms10050864

**Published:** 2022-04-21

**Authors:** Eun Jeong Won, Hyun Hee Jang, Hansoo Park, Seong Jin Kim

**Affiliations:** 1Department of Laboratory Medicine, Chonnam National University Hwasun Hospital, Hwasun 58128, Jeollanam-do, Korea; parasite.woni@jnu.ac.kr; 2Department of Parasitology and Tropical Medicine, Chonnam National University Medical School, Hwasun 58128, Jeollanam-do, Korea; jang990303@naver.com; 3Department of Biomedical Science and Engineering, Gwangju Institute of Science and Technology (GIST), Gwangju 61005, Korea; hspark27@gist.ac.kr; 4Department of Dermatology, Chonnam National University Hospital, Gwangju 61469, Korea

**Keywords:** scalp microbiome, alopecia areata, prognosis, *Staphylococcus caprae*, *Corynebacterium*, *Cutibacterium* species

## Abstract

Little is known about the scalp bacterial composition of alopecia areata (AA) patients. The aim of this study was to investigate the differences in the scalp microbiome of AA patients according to their prognosis, in addition to healthy controls. A total of 33 AA patients and 12 healthy controls (HC) were included in this study. The microbiomes were characterized by sequencing 16S rRNA genes on the Illumina MiSeq platform. The scalp microbiome was more diverse in AA patients compared to HC, but not significantly different according to the severity of AA. Nevertheless, the higher proportion of *Corynebacterium* species and the lower proportion of *Staphylococcus caprae* among the *Staphylococcus* species were noticed in severe AA patients compared to HC or mild AA. The higher ratio of *Cutibacterium* species to *S. caprae* was noticed in severe AA. We highlight the potential predictive role of scalp microbiome profiling to a worse prognosis of patients with alopecia areata.

## 1. Introduction

Alopecia areata (AA) is the second most common type of hair loss disorder in humans, which affects nearly 2% of the general population [1]. The most common type is a small annular or patchy bald lesion usually on the scalp [2], and these can extend to the entire scalp (Alopecia totalis) or to the entire pilar area of the body (Alopecia universalis). This disorder can have a devastating impact on the shaping of the body’s image and self-esteem, causing, in many cases, inefficient functioning in the social environment and eventually decreased quality of life [3]. However, the management of AA still remains a challenge. Several speculations on the causes of AA have been proposed, such as loss of immune privilege/immune dysregulation, genetic predisposition, hormonal imbalance, emotional or physical stress, genetic tendencies, and other local skin disorders and nutritional deficiencies [4]. Among the proposed causes, a link with the gut microbiome has also been hypothesized [5]. *Cutibacterium*, *Staphylococcus*, and *Corynebacterium* were the most common bacteria, while *Malassezia* was the most common fungus, on healthy scalps [6,7]. Due to its unique features, the scalp is expected to harbor a specific microbiome, which is expected to play a peculiar role in scalp conditions related to hair growth [7]. Several researchers studied the scalp microbiome profile regarding AA and they suggested the notion that an imbalance between skin microbes that maintain skin homeostasis causes inflammation [8,9,10,11,12]. Several microbes such as *Cutibacterium acnes*, *Staphylococcus*, and *Malassezia* were suggested to be implicated in AA, but it still needs to be clarified using various cohorts of different populations. Here, we aimed to investigate scalp microbiome profiling in patients with alopecia areata and to assess their prognostic implications.

## 2. Materials and Methods

This study was carried out in accordance with all relevant institutional guidelines. The Ethics Committee of Chonnam National University Hospital approved this study (CNUH-2020-118) and written informed consent was obtained from all subjects.

All patients with AA were evaluated at a tertiary university hospital between May 2020 and February 2021. A total of 33 AA patients and 12 healthy controls were finally included in this study. Clinical and demographic information for each patient was recorded, including age, gender, age of disease onset, duration of current episode of disease, and any underlying immunological disorders. The severity of AA was evaluated based on the extent of scalp hair loss, progression of disease, and response of treatment, and was divided into mild AA (*n* = 26) and severe AA (*n* = 7) (Table 1). The scalp samples were obtained by means of swab procedure according to previously reported methods [8,9,10] with minor modifications. Sterile cotton swabs were soaked for at least 30 s in solution (0.9% NaCl and 0.1% Tween 20). After collection, the head of each swab was cut and stored in solution. The samples were transferred to our laboratory and stored at −80 °C until DNA extraction. Bacterial DNA from scalp swabs was extracted by means of the DNeasy Blood&Tissue kit (Qiagen, Hilden, Germany) according to manufacturer protocol, with minor modifications [11,12,13]. The DNA samples were amplified using 16S universal primers that target the V3–V4 region of the bacterial 16S ribosomal gene. Detection of the sequencing fragments was performed using Illumina MiSeq technology by Macrogen (Seoul, Korea). The sequences were trimmed and merged, and then clustered into operational taxonomic units (OTUs) using the CLC Genomics Workbench v. 10.1.1 and CLC Microbial Genomics Module v. 2.5 (Qiagen, Hilden, Germany). Taxonomic assignment of these sequences was carried out based on the NCBI taxonomy database, with an OTU cutoff of 3%. The most abundant sequences were considered representative of each cluster and assigned to a taxonomy level based CLC Microbial Genomics default values. A multiple sequence alignment with MUSCLE v.3.8.31 was performed to obtain an OTU table together with the pre-aligned 16S and 18S data from the SILVA v132 database. Alpha diversity metrics (richness and Total number) were calculated based on rarefied OTU counts and an exploratory analysis of beta-diversity (between-sample diversity) was performed based on the Bray-Curtis measure of dissimilarity as a principal coordinate analysis (PCoA). For the hierarchical cluster analysis, Bray-Curtis metrics and complete linkage clustering were implemented. Volcano plots were created to show estimated log2-fold differences in OTU abundance between AA and HC. A two-tailed Student’s *t* test and the χ^2^ test were used to compare the differences in scalp microbiomes between the groups using GraphPad Prism software (GraphPad Software Inc., San Diego, CA, USA). PERMANOVA analysis was carried out to ascertain the statistical significance of the clusters using the tools in the CLC Microbial Genomics Module v. 2.5 (Qiagen).

## 3. Results

A total of 3,692,142 good-quality reads with a mean length of 301 base pairs were generated. The results of the alpha diversity metrics (Shannon index) are shown in boxplots in Figure 1A. The Shannon index was significantly higher in the AA patients than in the HC (all groups, *p* < 0.001), reflecting a significantly higher species richness in the former group. In contrast, there was no significant difference in the alpha diversity according to the severity of AA. The unweighted beta diversity analysis showed the overall bacterial community structure and phylogenetic diversity between AA patients and HC (Figure 1B,C). Overall, we could not find any distinct composition of the scalp microbiome specific to AA or its severity. The statistical values obtained by the PERMANOVA test were not significant (AA vs. HC, *p* = 0.29543; mild AA vs. severe AA, *p* = 0.0825). The bacterial compositions of the scalp samples were further examined. At the family level, Staphylococcaceae and Burkholderiaceae were more abundant in HC and mild AA than in severe AA (Figure 2A). The proportion of Propionibacteriaceae increased in mild AA but not in severe AA. At the genus level, severe AA exhibited a lesser abundance of *Staphylococcus* species and *Ralstonia* species (Figure 2B). The volcano plot also presented that the genus *Corynebacterium* that is represented in the microbiome by the several OTU: *Corynebacterium ihumii*, *Corynebacterium simulans*, *Corynebacterium* sp. NML98-0116, and *Corynebacterium* sp. KPL 1996 were more distinctive to AA than HC (Appendix A). Something of interest that we found was that the proportion of *S. caprae* among the *Staphylococcus* species was markedly declined in severe AA (mean, 49.5% vs. 42.1% vs. 10.5%, HC vs. mild AA vs. severe AA) (Table 2), while the proportion of *Corynebacterium* species was significantly increased in severe AA (mean, 0.3% vs. 0.6% vs. 6.3%, HC vs. mild AA vs. severe AA). The mean ratio of *Cutibacterium* species to *S. caprae* was 0.97 for HC, 2.13 for mild AA, and 16.01 for severe AA, respectively.

## 4. Discussion

Several scientific published evidence have reported the strict correlation between microbial disequilibrium and skin conditions [4,5,7,8,9,10]. Given this growing body of literature, it is becoming increasingly clear that the modulation of the microbiota may be a novel and important adjunct modality. However, the knowledge regarding the scalp microbiota associated with the prognosis in AA patients is still lacking. Our data firstly showed that the certain composition of the scalp microbiota enabled us to predict the intractable AA patients. We found that a higher diversity of bacterial species inhabited the scalp of AA subjects, which is in line with previous studies [8,14]. Pinto et al. announced that this may be a piece of evidence that suggests that an unhealthy scalp is susceptible to being colonized by microorganisms [8]. They proposed that the higher *Propionibacterium acnes*/*S. epidermidis* ratio or *P. acnes*/*S. aureus* ratio were noticed in the AA subjects, however, they were not significant in our study. We also found that the proportion of Propionibacteriaceae was increased in mild AA compared to HC, but not in severe AA. The role of *Propionibacterium* species, especially *Propionibacterium acnes*, may differ in the stage of AA, or the specific region of hair follicle even in the same stage [9]; thus, it still needs to be further clarified.

In this study, we noticed that the potential role of *S. caprae* among the *Staphylococcus* species harbored a negative correlation for the disease progression of AA. Currently, *S. caprae* has been recognized as a commensal coagulase-negative *staphylococcus* that usually colonizes the nose, nails, and skin [15], which can cause community-acquired and/or hospital-acquired infections in humans. We could speculate that *S. caprae* is likely to be colonized in healthy conditions of the scalp, although this notion should be demonstrated by further experiments. Collectively, we could suggest that the high *Cutibacterium* species/*S. caprae* ratio can be useful to predict severe AA, which is similar to the *Propionibacterium acnes*/*S. epidermidis* ratio previously announced.

In addition to *Staphylococcaceae*, *Corynebacteriaceae* is known to be the one of the major skin microbiomes [6,7]. Dimitriu et al. found that *Corynebacterium* OTUs were associated with chronological age and skin aging on the forehead [16]. We also found that the proportion of *Corynebacterium* species was higher in severe AA than in mild AA or HC. This is suggestive of the potential to predict a worse prognosis of AA. It is unclear yet how certain *Corynebacterium* species can colonize with dominance in severe AA and whether the effect of *Corynebacterium* species can be reversible with proper treatment or not. Of the dominant species on the skin surface, *Burkholderia* spp. were reported to be predominantly colonized in hair follicles in androgenic alopecia patients [9]. Ho et al. did not find an obvious correlation with androgenic alopecia severity, region, or hair follicle compartment for the distribution of different species. On the other hand, we observed that *Burkholderiaceae* or *Ralstonia* species were found to be more abundant in HC and mild AA than in severe AA. The protective effect of these species is suggestive but should be further clarified using a larger number of cohorts.

Our study had several limitations. In particular, we tried to elucidate the role of the scalp microbiome using 16S rRNA amplicon sequencing approaches, which is most commonly used for the study of microbiome composition. It provided reliable classification down to the genus level, but the limited discriminatory power for certain genera at the species level has been noticed [17]. We also could not confirm which *Corynebacterium* species was the key factor to the prognosis of AA exactly. Further scrutiny is warranted requiring confirmation through additional tests such as cultivation, real-time PCR, and long-read sequencing, which will confidently confirm the presence or absence of bacterial species [18]. In addition, further assessment may be necessary to clarify the relevance of *Vibrio* ambiguous taxa, which was unexpectedly observed among the scalp microbiome in either HC or AA. This study is provisional but supports that scalp microbiome profiling can predict the prognosis of patients with alopecia areata. Our dataset is too small to be representative of a true population subset and needs further longitudinal studies that enroll larger numbers of subjects with diverse populations. Regretfully, all the scalp samples were not collected at the first diagnosis, and this makes the prognostic value of the scalp microbiome weakened. Regardless, our data could serve as the basis for biomarkers related to the scalp microbiome and pave the way for an in-depth understanding of their possible role to improve patient-care plans in AA patients.

## Figures and Tables

**Figure 1 microorganisms-10-00864-f001:**
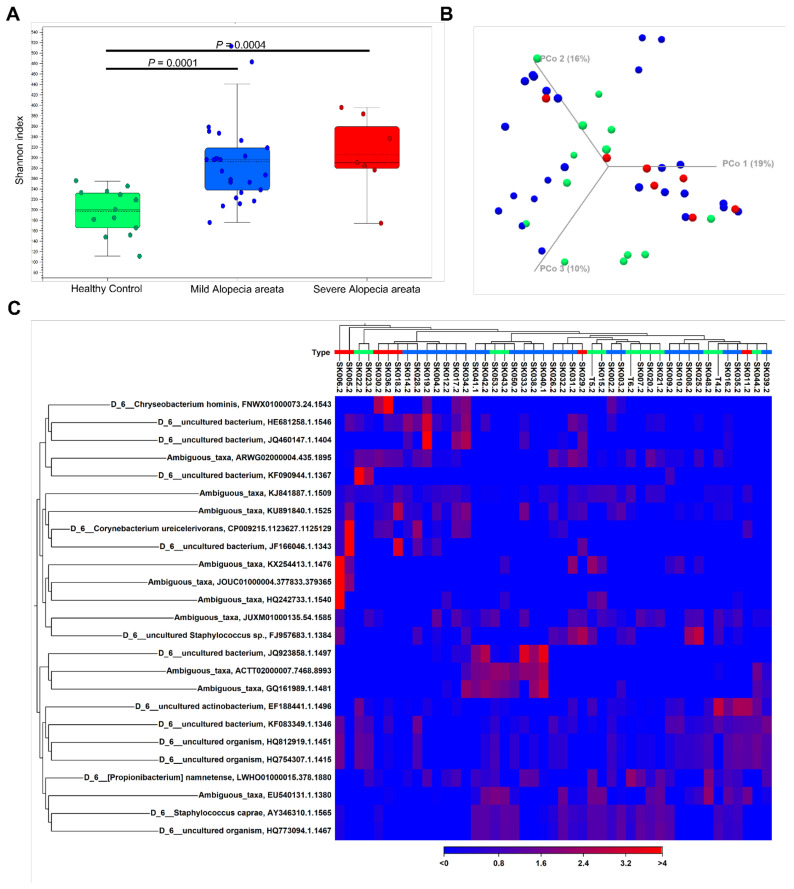
The alpha diversity and beta diversity indices of scalp microbiome between alopecia areata (AA) with different prognosis and healthy control (HC). (**A**) The significantly higher Shannon index was found in the AA patients than in the HC. There was no significant difference in the alpha diversity according to the prognosis among the AA patients. Statistical analysis was performed by a Mann–Whitney U-test. Plotted are interquartile ranges (IQRs; boxes), medians and means (dark lines and dotted lines, respectively, in the boxes), and the lowest and highest values within 1.5-fold of the IQR of the first and third quartiles (whiskers above and below boxes). (**B**) Unweighted beta diversity analysis showed the overall bacterial community structure and phylogenetic diversity of scalp microbiome between mild AA (blue dots), severe AA (red dots), and HC (green dots). (**C**) Heat map of OTU abundances of the scalp microbiome and the unweighted beta diversity analysis: healthy control (green), mild AA (blue), and severe AA (red).

**Figure 2 microorganisms-10-00864-f002:**
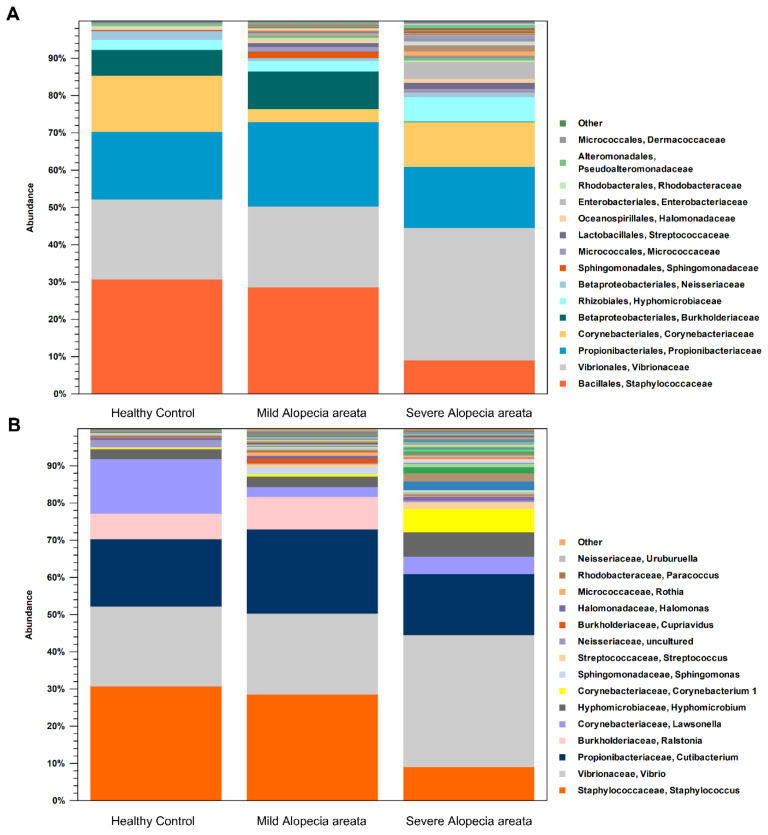
Composition of the scalp microbiome in AA patients (**A**) at the Family level, Staphylococcoceae was significantly reduced among scalp microbiome in severe AA. (**B**) At the Genus level, *Staphylococcus* species was significantly reduced but *Corynebacterium* species was markedly increased in severe AA.

**Table 1 microorganisms-10-00864-t001:** Clinical parameters of study cohort involved in this study.

	Mild Alopecia Areata (*N* = 26)	Severe Alopecia Areata (*N* = 7) *
Male (%)	5 (71.4)	17 (65.4)
Mean age	19.5 (10–43)	33.9 (9–60)
Ranges of hair loss involvement (%)	5–85%	95–100%
Average of hair loss involvement (%)	40.0	98.3
Progressive hair loss or non-response to therapy, No. (%)	7 (26.9)	7 (100.0)
Size increment, No. (%)	17 (65.4)	4 (57.1)
Spreading tendency, No. (%)	19 (73.1)	3 (42.9)
Hair growth, No. (%)	18 (69.2)	2 (28.6)
Median duration (Day)	151.5	746
Presence of autoimmune disorders, No. (%)	3 (11.5)	3 (42.9)

* Alopecia totalis was designated into severe alopecia areata.

**Table 2 microorganisms-10-00864-t002:** Potential prognostic markers associated with the scalp microbiota of alopecia areata (AA) patients.

	HC	Mild AA	Severe AA
*Staphylococcus caprae*/*Staphylococcus* species	49.5% (115,707/233,865) ^a^	42.1% (176,043/418,857)	10.5% (3739/35,653) ^a^
*Corynebacterium* species/Total	0.3% (2307/698,287) ^b^	0.6% (8860/443,367) ^a^	6.3% (23,134/116,344) ^a,b^
*Cutibacterium* species/*S. caprae*	0.97 (113,005/115,707)	2.13 (375,500/176,043)	16.01 (59,877/3739)

^a^ Statistical difference (*p* < 0.05) was noticed between the two groups. ^b^ Statistical difference (*p* < 0.01) was noticed between the two groups.

## Data Availability

Data sharing not applicable as no datasets generated and/or analyzed for this study. Data are available upon reasonable request. All data relevant to the study are included in the article or uploaded as Appendix A.

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
