# Peer review of "A Potential Predictive Role of the Scalp Microbiome Profiling in Patients with Alopecia Areata: Staphylococcus caprae, Corynebacterium, and Cutibacterium Species"

_microorganisms, 2022, doi:10.3390/microorganisms10050864_

Round 1
Reviewer 1 Report
Dr. Riley Wu,
Assistant Editor Microorganisms
Dear Dr. Wu,
The Manuscript ID: microorganisms-1672440 entitled “A potential predictive role for metagenomic profiling of the scalp microbiome in patients with alopecia areata: Staphylococcus caprae, Corynebacterium, and Cutibacterium species” is a well written manuscript describing the microbiome of the scalp from patients with alopecia areata. The manuscript is scientifically sound and the use of metagenomic as a method of species identification is promising. I really enjoyed reading this manuscript and I believe that it is of interest of Microorganisms’ public. It would be interesting if the manuscript could take the bellow’s comments before be published in the Microorganisms’ Journal.
Specific comments:
1- Table 1: There are two numbers (19 and 18) with their respective percentages refereeing to “Spreading tendency”. Please, explain which one should be used.
2- In the sentence “A multiple sequence alignment with MUSCLE v.3.8.31 was performed to obtain an OTU table, together with the prealigned 16S and 18s data….”, the “s” from 18S should be capitalized.
3- In the sentence “While, the proportion of Corynebacterium 1 species was significantly increased in severe AA….” and in the legend of the Figure 2, the “1” after Corynebacterium seems to be out of place. Is this correct?
4- The legends of the Figure 2 are too small. It is possible to increase them?
Reviewer 2 Report
- Title. Should be changed. The authors did not analyzed metagenome, but only bacteriome, using 16SrDNA amplicon sequencing
- Abstract. Please define abbreviation HC on first use in abstract
- Abstract and Intoduction. What is it “metagenomics profiling”? The authors did not analyzed metagenome.
- 18s - capital letter required
- Two-tailed Student’s t test, the χ2 test, and the PERMANOVA test were used to test for differences in the phenotypical characteristics of the microbiome using GraphPad Prism software (GraphPad Software Inc., San Diego, CA, USA). -- incomprehensible sentence
GraphPad is the software for the comprehensive analysis and powerful statistics, and for the creation of the graphs. The authors used GraphPad for the two-tailed Student’s t test and the χ2 test. The PERMANOVA test is available in the CLC. And what means “phenotypical characteristics”?
- Figure 1A - you need to sign the Y axis. Figure 1 A, C - The font of the inscriptions needs to be increased
- Figure 2 - The font of the inscriptions needs to be increased
- Results and Supplementary Figure 1. The authors can not mention the Corynebacterium species, Propionibacterium species, and so on, only OUT, because they analyzed 16S rDNA sequences
- What controls did the authors use to prove that the presence of Burkholderia is not a laboratory contamination?
- The authors had 3,692,142 good-quality reads. And how many reads they had for one HC and one AA sample?
- Why the reference 13 is not mentioned in the Introduction?
Reviewer 3 Report
This manuscript describes a metagenomic analysis of the scalp microbiome in patients with alopecia areata. The analysis focusses on Staphylococcus caprae, Corynebacterium, and Cutibacterium species. The results showed a higher ratio of Cutibacterium species to S. caprae in patients with severe alopecia areata. The authors discussed the potential predictive role of metagenomics profiling of the scalp microbiome to the worse prognosis of patients with alopecia areata. The work is interesting and fits the scope of the journal. However, there are a few points which need to be addressed by the authors:
1) The authors employed metagenomics analysis of DNA samples based on 16S universal primers targeting the V3–V4 region of the bacterial 16S ribosomal gene. The authors should discuss the accuracy of their analysis and how do they discriminate species of high similarity. Why do authors not use other approach to sequence additional variable regions for improving the accuracy ?
2) The resolution of figure 1b and 1c should be improved – it is hardly readable.
3) The resolution of figure 2 (a,b,c) is very low and they are not readable at all.
4) The dataset that have been used are not enough for making a clear conclusion and provide evidences. How do authors plan to improve their study in the future ?
Round 2
Reviewer 2 Report
Dear Authors!
The text of the manuscript and figures need to be corrected.
A potential predictive role of the scalp microbiome profiling in patients with alopecia areata: Staphylococcus caprae, Corynebacterium, and Cutibacterium species
Eun Jeong Won1,2, Hyun Hee Jang2, Hansoo Park3 and Seong Jin Kim4
- The authors include Staphylococcus caprae in the title of the manuscript, without explaining in the text of the manuscript how they distinguished this species from other species of coagulase-negative staphylococci using only a V3–V4 region of the 16S rDNA Should be explain.
- The authors compare the scalp microbiome of healthy and sick people, but in the Introduction, they do not provide literature data on the composition of a healthy scalp microbiome. Need to add.
- Figure 2A and 2B – the figure captions (y-axis values, description of taxa) should be improved and made readable
- Figure 2C - should be improved: the font should be enlarged, the abbreviations should be deciphered
- All revealed taxa should be listed and discussed, including the Vibrio, which is present in an unexpectedly high percentage even in healthy people (Figure 2)
- Volcano plot also presented that certain Corynebacterium species such as Corynebacterium ihumii, Corynebacterium simulans, Corynebacterium sp. NML98-0116, Corynebacterium sp. KPL 1996. Should be changed on “the genus Corynebacterium is represented in the microbiome by the several OUT: Corynebacterium ihumii, Corynebacterium simulans, Corynebacterium sp. NML98-0116, Corynebacterium sp. KPL 1996”.
Reviewer 3 Report
The authors have suitably revised the article, which has been imrpoved.
Author Response
Thank you for your kind comments.